# The Marine Antimicrobial Peptide AOD with Intact Disulfide Bonds Has Remarkable Antibacterial and Anti-Biofilm Activity

**DOI:** 10.3390/md22100463

**Published:** 2024-10-08

**Authors:** Ruoyu Mao, Qingyi Zhao, Haiqiang Lu, Na Yang, Yuanyuan Li, Da Teng, Ya Hao, Xinxi Gu, Jianhua Wang

**Affiliations:** 1Gene Engineering Laboratory, Feed Research Institute, Chinese Academy of Agricultural Sciences, Beijing 100081, China; 2Innovative Team of Antimicrobial Peptides and Alternatives to Antibiotics, Feed Research Institute, Chinese Academy of Agricultural Sciences, Beijing 100081, China; 3Key Laboratory of Feed Biotechnology, Ministry of Agriculture and Rural Affairs, Beijing 100081, China; 4Enzyme Engineering Laboratory, College of Food Science and Technology, Hebei Agricultural University, Baoding 071001, China

**Keywords:** antimicrobial peptides, American Oyster Defensin (AOD), disulfide bond, anti-biofilm activity

## Abstract

American Oyster Defensin (AOD) is a marine peptide that is derived from North American mussels. It has been demonstrated to exhibit potent antimicrobial activity and high safety in both in vitro and in vivo models. In this study, to facilitate synthesis, mutants of AOD with fewer disulfide bonds were designed and subjected to structural, antimicrobial, and anti-biofilm analysis. The antimicrobial activity of AOD-derived peptides decreased after reduction in the disulfide bond, and among its three derivatives, only AOD-1 inhibited very few bacteria with a MIC value of 64 μg/mL, whereas the others had no inhibitory effect on pathogenic bacteria. The findings demonstrated that full disulfide bonds are indispensable for bactericidal activity, with the α-helix playing a pivotal role in inhibiting bacterial membranes. Furthermore, the results of the ATP, ROS, membrane potential, and membrane fluidity assays demonstrated that intracellular ATP, reactive oxygen species, and membrane fluidity were all increased, while membrane potential was reduced. This indicated that AOD resulted in the impairment of membrane fluidity and induced metabolic disorders, ultimately leading to bacterial death. The inhibitory effect of AOD on the biofilm of *S. epidermidis* G-81 was determined through the crystal violet and confocal microscopy. The results demonstrated that AOD exhibited a notable inhibitory impact on the biofilm of *S. epidermidis* G-81. The minimum biofilm inhibitory concentration of AOD on *S. epidermidis* G-81 was 16 μg/mL, and the minimum biofilm scavenging concentration was 32 μg/mL, which exhibited superior efficacy compared to that of lincomycin. The inhibitory effect on the primary biofilm was 90.3%, and that on the mature biofilm was 82.85%, with a dose-dependent inhibition effect. Concurrently, AOD cleared intra-biofilm organisms and reduced the number of biofilm-holding bacteria by six orders of magnitude. These data indicate that disulfide bonds are essential to the structure and activity of AOD, and AOD may potentially become an effective dual-action antimicrobial and anti-biofilm agent.

## 1. Introduction

*Staphylococcus epidermidis* (*S. epidermidis*) is a coagulase-negative staphylococcus that colonizes mucous membranes or skin surfaces [1]. In contrast to the acute symptoms of *Staphylococcus aureus* infections, *S. epidermidis* infections typically manifest as chronic conditions, and optimal treatment may be overlooked due to the absence of clinical symptoms. Generally considered non-pathogenic, an *S. epidermidis* infection only presents with symptoms when there is an imbalance in the host immune system or when external medical devices are retained [2]. The pathogenic mechanism primarily involves the formation of a bacterial biofilm through adhesion. In clinical practice, antibiotics are commonly used for treatment; however, the presence of bacteria within the biofilm makes them less susceptible to antibiotic removal, resulting in recurrent infections [3]. Therefore, the presence of *Staphylococcus epidermidis* infections necessitates the development of a novel therapeutic approach in an urgent manner.

Biofilm is a complex, three-dimensional biological community formed when bacteria secrete polysaccharides, proteins, and biological fibers as they proliferate. Within this process, an abundance of exopolysaccharides, proteins, and DNA are produced to construct the matrix of the biofilm while creating a microenvironment that sustains bacterial activity within the membrane [4]. Bacteria residing within the biofilm exhibit 10–1000 times greater resistance to drugs compared to free-floating bacteria [5]. In the later stages of biofilm development, inadequate nutrient supply leads to its breakdown and subsequent release of free bacteria that reattach onto host surfaces, causing recurrent infections [6].

Currently, antibiotics are the primary approach for managing clinical infections in the majority of cases. However, the emergence of drug-resistant bacteria has necessitated an urgent requirement for the development and implementation of alternative antibiotic solutions [3]. Antimicrobial peptides (AMPs) play a pivotal role in the innate immune response, exhibiting a diverse range of antibacterial, immunomodulatory, antitumor, and other therapeutic effects. Consequently, they offer a promising alternative to antibiotics as a first-line treatment option. American Oyster Defensin (AOD) is a marine peptide originating from North American mussels, comprising 38 amino acids and featuring three pairs of disulfide bonds (C4-C25, C11-C33, and C15-C35) [7]. The expression and purification of AOD in *Pichia pastoris* X-33 have been successfully achieved [8]. The recombinant AOD exhibits potent bactericidal activity against *Staphylococcus aureus*, *Streptococcus agalactiae*, *S. epidermidis*, and other pathogens. Moreover, AOD demonstrates low toxicity levels and high stability both in vivo and in vitro. The Minimum Inhibitory Concentration (MIC) of AOD against clinical strains of *S. epidermidis* G-81 was determined to be 4 μg/mL. In vitro bactericidal kinetics analysis reveals that the bactericidal rate of *S. epidermidis* G-81 reaches 99.9% within 0.5 h upon treatment with 2× and 4× MIC concentrations, which remains constant for 24 h without any rebound effect observed [8].

The disulfide bond is a common and essential structural element in defensins, playing a key role in their antibacterial activity, proper folding, and structural stability [9]. The investigation of the structure–activity relationship between disulfide bonds and defensins, as well as the mechanism underlying pathogen eradication, has emerged as a prominent research focus in this field. It has been reported that coprisin from *Copris tripartitus*, consisting of 43 amino acids, lacked any disulfide bond pairs, thereby attenuating its antibacterial activity while still retaining its antifungal activity [10]. The linear hBD-3-l has been demonstrated to exhibit the highest activity, whereas the stability of hBD-3 against protease activity diminishes as the number of disulfide bonds decreases [11].

In this study, we investigated the impact of AOD’s disulfide bond on the secondary structure and its bactericidal efficacy, as well as examined the in vivo safety of AOD and its co-administration effects. Additionally, we explored the mechanism underlying AOD’s bactericidal effect against *S. epidermidis* and its inhibitory effect on biofilm formation.

## 2. Results

### 2.1. AOD Mutants Design Based on Disulfide Bonds

To evaluate the effect of disulfide linkage on the activity of AOD, the derivatives were designed: AOD-1 (C15, C35→S15, and S35), AOD-2 (C4, C25→S4, and S25), and AOD-3 (C11, C33→S11, and S33). The cysteines were exchanged with serines because they proved to be an attractive substitutable residue with a similar chemical composition and structure compared to cysteines [12]. As shown in Table 1, the molecular weight of derivatives was highly similar to the parent peptide AOD, whereas AOD-1, AOD-2, and AOD-3 had a higher isoelectric point and net charge and a lower GRAVY, which indicated the derivatives were more hydrophilic.

### 2.2. MIC of AOD and Its Derivatives

The MIC values of AOD and its derivatives are shown in Table 2. AOD had a potent bactericidal effect on most of the Gram-positive bacteria, such as *Staphylococcus epidermidis* and *Staphylococcus aureus*, with MIC values of 4–16 μg/mL. Its derivative, AOD-1, only inhibited very few of the bacteria, with a MIC value of 64 μg/mL, while the other derivatives demonstrated no inhibitory effect against the pathogens, with MIC values exceeding 64 μg/mL. This suggests that the disulfide bonds are crucial for the bactericidal activity of AOD.

### 2.3. CD Spectroscopy of AOD and Its Derivatives

CD spectroscopy was used to measure the secondary structure of AOD in the presence of H_2_O (to simulate water), 50% TFE, and 20 mM SDS (to simulate microbial membranes), respectively. It has been shown that AOD has the α and β structure in all three environments, as evidenced by two negative peaks at wavelengths of approximately 208 nm and 222 nm (Figure 1). The CD peaks for AOD were reduced in H_2_O and 50% TFE but increased in 20 mM SDS, suggesting that it tends to form an α-helix structure under microbial membrane conditions. The main structure in which the antimicrobial peptide functions is the α-helix. In the same environment, the α-helix content of the derivatives of AOD was reduced (Table 3) and their bactericidal activity subsequently deteriorated, and it was speculated that it might be due to the change in the position and number of disulfide bonds that resulted in less α-helix content in the two-sister structure and lower bactericidal activity. This suggests that the α-helix has a very important role in the process of AOD functioning.

### 2.4. Synergy with Antibiotics

The effect of the action of AOD in combination with other drugs was judged by the combination index (fractional inhibitory concentration index, FICI). Table 4 shows the bactericidal effect of AOD combined with four commonly used bactericidal drugs (cefotaxime sodium, tetracycline, vancomycin, and ciprofloxacin) on *S. epidermidis* G-81. The FICI values of AOD and cefotaxime, tetracycline, ciprofloxacin, and vancomycin were 0.575, 1, 1.125, and 0.581, respectively. The combined effects of AOD and cefotaxime, as well as tetracycline and vancomycin, were additive. Meanwhile, AOD and ciprofloxacin had independent effects, and no antagonistic effects were observed.

### 2.5. The Safety of AOD In Vivo

The in vivo safety assessment of AOD revealed no discernible stress response or abnormal behavior in the mice throughout the 7-day experimental period, and there was no observed weight loss (Figure 2A). The kidneys and livers of mice were collected one week after intraperitoneal injection, and tissue sections were prepared for HE staining (hematoxylin-eosin staining), as depicted in Figure 2B. Notably, no histomorphological abnormalities were observed in the tissues, indicating the absence of any detrimental effects of AOD on these organs. Furthermore, there were no significant differences in whole-cell profiles or serum biochemical indexes between the experimental group and control group, suggesting normal metabolic function in the mice (Table 5). Collectively, these findings affirm that AOD is a safe candidate drug. 

### 2.6. Fluorescence Staining of S. epidermidis G-81

The fluorescence staining of *S. epidermidis* G-81 after treatment is depicted in Figure 3, revealing a conspicuous absence of red-colored bacteria in the untreated control group, indicating negligible cell death. The red-colored bacteria were predominantly observed in the 1× and 2× MIC AOD-treated groups, and no green-colored bacteria were detected in the 4× MIC-treated group, indicating a near-complete eradication of bacteria. The fluorescence staining results indicate that AOD has a destructive effect on the cell membrane, leading to the rupture of the cell membrane and the exposure of DNA.

### 2.7. Effect of AOD on Cell Membrane Fluidity in S. epidermidis G-81

The GP value is an index that reflects the overall fluidity of the cell membrane. A change in the GP value indicates a change in membrane fluidity. As shown in Figure 4A, the GP value of the control was 0.206. Following treatments with 1×, 2×, and 4× MIC AOD as well as 2× MIC lincomycin, the GP value of *S. epidermidis* G-81 exhibited a decline. Notably, this decline was more pronounced at concentrations of 2× and 4× MIC AOD, measuring 0.283 and 0.294, respectively. These findings suggest that AOD induces membrane stiffening and impairs fluidity. Meanwhile, the 2× MIC lincomycin did not change it significantly.

### 2.8. Intracellular ATP Assay

The intracellular ATP of the bacteria reflected the metabolism of the bacteria. The intracellular ATP levels of *S. epidermidis* G-81 were significantly increased by 3.58-, 5.94-, and 6.12-fold after treatment with AOD at 1×, 2×, and 4× MIC concentrations, respectively (Figure 4B). Moreover, the intracellular ATP exhibited a substantial variation in response to increasing AOD concentration, indicating that AOD disrupts bacterial intracellular metabolism and leads to excessive ATP production. 

### 2.9. Effect of AOD on Reactive Oxygen Species in S. epidermidis G-81

Dichlorofluorescein diacetate (DCFH-DA) is a reactive oxygen species detection reagent. As shown in Figure 4C, the AU values of *S. epidermidis* G-81 were 303.25, 345.25, 455.25, and 349.5 after treatment with 1×, 2×, and 4× MIC AOD and 2× MIC of lincomycin, respectively. The findings suggest that AOD triggers the generation of reactive oxygen species in *S. epidermidis* G-81 cells, resulting in enhanced cellular apoptosis and diminished development of drug resistance [13].

### 2.10. MBIC/MBEC of AOD on S. epidermidis G-81

Minimum biofilm inhibitory concentration (MBIC) and minimum biofilm eradication concentration (MBEC) are the lowest concentrations that inhibit the initial formation of biofilm and remove biofilm, respectively. The MBIC of AOD against *S. epidermidis* G-81 was 16 μg/mL, while the MBEC was 32 μg/mL (Table 6). These values were significantly superior to those of the control lincomycin, with an MBIC of 64 μg/mL and an MBEC exceeding 128 μg/mL. 

### 2.11. Effect of AOD on Primary and Mature Biofilms of S. epidermidis G-81

The inhibitory effect of AOD on biofilm formation was evident during the early stages, as depicted in Figure 5A. At concentrations of 1× MIC (4 μg/mL), 2× MIC (8 μg/mL), and 4× MIC (16 μg/mL), the ability to inhibit biofilm formation was measured at 27.75%, 30.66%, and 55%, respectively. Notably, at a concentration of 32× MIC (128 μg/mL), the inhibitory efficacy reached an impressive level of 90.3%.

The inhibitory effect of AOD on mature biofilm was demonstrated in Figure 5B, exhibiting inhibitory rates of 29.06%, 31.33%, and 45.66% at concentrations equivalent to the minimum inhibitory concentration (MIC) of 1× MIC (4 μg/mL), 2× MIC (8 μg/mL), and 4× MIC (16 μg/mL), respectively. At a concentration of 32× MIC (128 μg/mL), AOD inhibited approximately 82.85% of the mature biofilm growth.

### 2.12. Biofilm Observation by Confocal Laser Scanning Microscopy

The untreated *S. epidermidis* G-81 (CK) exhibited a primary biofilm thickness of 37.5 μm during the early stage of biofilm formation (Figure 6). Following treatment with AOD concentrations of 1× MIC, 2× MIC, and 4× MIC, the biofilm thickness decreased to 6 μm, 12.5 μm, and 6.5 μm, respectively. The dose-relationship of AOD to the thickness of the primary biofilm was not significant, but the density of bacteria in the biofilm decreased with the increase in the concentration. These findings demonstrate that AOD has a substantial impact on inhibiting and preventing initial-stage biofilm formation. 

At the late stage of biofilm formation, the mature biofilm thickness of untreated *S. epidermidis* G-81 (CK) reached 36 μm. After treatments with 1× MIC, 2× MIC, and 4× MIC concentrations, a decreasing trend in biofilm thickness was observed, measuring 21 μm, 9 μm, and 7 μm, respectively. Additionally, there was a significant reduction in bacterial density. The results indicated that AOD had a dose-dependent effect on eliminating biofilm at the late stage of biofilm formation.

### 2.13. Effect of AOD on the Biofilm-Retaining Bacteria S. epidermidis G-81

The effect of AOD on persister bacteria in mature biofilms was further evaluated by introducing a high dose of vancomycin (100× MIC) to induce biofilm persister. The dose-dependent effect of AOD on the removal of persistent bacteria is evident in Figure 7. Following incubation with AOD at concentrations of 4, 16, and 128 μg/mL, a remarkable reduction in intramembrane persistent bacteria was observed by 4, 5, and 6 orders of magnitude, respectively. This efficacy surpasses that of the control lincomycin.

## 3. Discussion

The physicochemical properties of antimicrobial peptides primarily encompass the assessment of their stability in vitro and in vivo, as well as the investigation into the structure–function relationship. Positive charge and hydrophilicity are the two main factors required for the antimicrobial activity of AMPs [14]. The GRAVY is the grand average of hydropathicity. AMPs bind to the surface of bacteria through electrostatic bonding, which is thought to be the first step in promoting the interaction between AMPs and cell membranes. In this work, the GRAVY index of AOD-1, -2, and -3 decreased from −0.363 to −0.573, indicating that the derivatives were more hydrophilic, which reduced their binding to the surface of the bacterial cell membrane and resulted in a decrease in activity.

Circular dichroism is an important method for determining the conformation of proteins or peptides in solution, which relies on measuring the differential absorption of left- and right-handed polarized light at specific wavelengths [15]. The SDS solution is used to mimic the hydrophobic environments of bacterial cell membranes [16]. TFE reduces the polarity of solvents and is often employed to simulate mammalian or bacterial cell membranes. The conformational changes in fungal defensin-like peptide P2 in SDS solution and 50% TFE solution were modeled. The findings revealed that P2 exhibited the highest α-helix content in SDS solution, whereas the α-helix content decreased in 50% TFE solution [17]. The plectasin-derived peptide NZ2114 exhibits comparable characteristics [18]. The AOD exhibiting complete three-pair disulfide bonds demonstrated the highest level of antimicrobial activity and exhibited the greatest abundance of α-helical structures in the TFE solution. The substitutions of cysteine with serine in the mutants lead to a substantial decrease in their antibacterial activity. Simultaneously, a concurrent decline is observed in the proportion of α-helix formation when exposed to SDS and TFE solutions (Figure 1 and Table 3). The substitution of cysteine as serine, therefore, exerts a profound influence on both the antibacterial efficacy and structural characteristics of AOD. Václav Čeřovský also discovered that the presence of three disulfide bonds enables Lucifensin, an antimicrobial peptide consisting of 40 amino acids, to maintain its specific conformation essential for exerting antimicrobial activity [19]. It was also shown that the replacement of cysteine by alpha-amino-n-butyric acid (Abu) residues in antimicrobial peptides Ar-1-Abu, Ar-2-Abu, and Ar-3-Abu resulted in the total loss of their antibacterial activity against the *Mycobacterium abscessus* variants [20]. This may be attributed to the fact that the preservation of intact disulfide bonds along with several hydrogen bonds in antimicrobial peptides is crucial for maintaining the requisite spatial structure necessary for their activity and stability [21,22]. The detailed information should be further studied in our next work. 

When developing antimicrobial peptides (AMPs) as therapeutics, it is of the utmost importance to evaluate their inhibitory activity under physiologically relevant conditions. Although a significant proportion of the AMPs currently in development meet the minimum inhibitory concentration (MIC) criteria in vitro, there are additional challenges to be overcome before they can be considered for therapeutic applications. When AOD was administered intraperitoneally for six consecutive days, the blood parameters and body weight of the mice were within the normal range, and the liver and kidney did not exhibit any lesions. This suggests that AOD does not affect the basal metabolism of the animals in vivo.

In our previous study, we observed the disruption of *S. epidermidis* G-81 cell membranes, leakage of contents, and irregularities in cell division using both scanning electron microscopy (SEM) and transmission electron microscopy (TEM) [8], which indicated that the bactericidal effect of AOD on *S. epidermidis* was most likely targeting the cell membranes. In this paper, the test of membrane fluidity also proved that AOD could make the bacterial cell membranes less fluid (Figure 4A). Concomitantly, intracellular ATP and reactive oxygen species demonstrated that AOD could disrupt the metabolism of *S. epidermidis* (Figure 4B,C), suggesting that the disruption of *S. epidermidis* G-81 by AOD is not limited to a single mechanism. This may be due to the disruption of the membrane affecting the metabolism within the bacteria. Additionally, Hao Ya et al. demonstrated that PN7 possesses multiple bactericidal mechanisms against *S. epidermidis* [23].

*S. epidermidis* is characterized by the presence of a multitude of biofilm-forming genes, which enable the bacterium to form a biofilm on any abiotic surface. Once a biofilm is established, its removal can be exceedingly challenging [24]. It can form a resistance barrier that continuously delivers resistant bacteria into the host while at the same time ensuring that the bacteria within the biofilm are in a low-energy and low-metabolism state, which makes it easy for resistance genes to form and transfer [25]. Some antibiotics have a certain ability to remove the biofilm, with erythromycin, rifampicin, and vancomycin demonstrating the most promising results and quinolone antibiotics exhibiting the strongest effect [26]. Nevertheless, the long-term usage of antibiotics can facilitate the development of bacterial resistance [27]. An antimicrobial peptide SCAMP extracted from snow crabs was found to be effective in reducing the formation of bacterial biofilms on submerged soft steel surfaces [28]. The membrane-active antimicrobial peptide 6K-F17 effectively inhibited biofilm production in *Pseudomonas aeruginosa* PAO1 and four multidrug-resistant (MDR) isolates from chronically infected CF individuals, as well as significantly reducing biofilm volume in both PAO1 and MDR isolates [29]. In this study, the potent inhibitory effect of AOD on the biofilm formation of *S. epidermidis* G-81 has been confirmed (Figure 6). Notably, AOD exhibited an even more superior inhibitory effect on the initial bacterial biofilm, achieving 90.3% inhibition at 32× MIC with a significant dose-dependent effect (Figure 5A). This inhibition may be attributed to the advantageous fast bactericidal effect of AOD, enabling rapid eradication of bacteria during the pre-biofilm formation stage. Meanwhile, AOD effectively suppressed biofilm growth by approximately 82.85% (Figure 5B). 

## 4. Materials and Methods

### 4.1. Strains, Peptides, and Reagents

The *S. aureus* ATCC 43300, *S. aureus* ATCC 25923, *S. epidermidis* ATCC 12228, and *E. coli* ATCC 25922 were purchased from the American Type Culture Collection (ATCC). *S*. *typhimurium* CVCC 14028 was purchased from the China Veterinary Culture Collection Center (CVCC). *S. flexneri* CMCC 51571 was purchased from the National Center for Medical Culture Collections (CMCC). *P. aeruginosa* CICC 21625 was purchased from the China Center of Industrial Culture Collection (CICC). *S*. *epidermidis* G-81 was presented by Professor Wu from China Agricultural University (Beijing, China). 

AOD pure product was prepared according to the previous methods [8] and stored in a refrigerator at −20 °C until use. The variants AOD-1, -2, and -3 were synthesized by WuXi AppTec (Shanghai, China) with a purity of over 92%. The molecular weight of the peptides was determined by electrospray ionization mass spectrometry (ESI-MS). Other parameters (isoelectric point, charges, and grand average of hydropathicity) were calculated by DBAASP (https://www.dbaasp.org/tools?page=property-calculation, 18 March 2023).

All other chemical reagents used in the experiment are analytically pure. Furthermore, six-week-old specific-pathogen-free (SPF) female BALB/c mice (approximately 20 g/mouse) were purchased from the Vital River Laboratories (VRL, Beijing, China). The tests regarding microbials were all carried out in Class II biological safety cabinets. All other chemical reagents used were analytical grade.

### 4.2. Physical and Chemical Properties

#### 4.2.1. MIC of AOD and Its Derivatives

The strains were incubated overnight at 37 °C and subsequently transferred to a fresh medium until they reached a logarithmic growth phase. Bacteria were then diluted to a concentration of 1 × 10^5^ CFU/mL and added to a 96-well plate, followed by sequential dilutions of peptides. Subsequently, the plate was incubated at 37 °C for 12–18 h. The minimum concentration where no bacterial growth occurred was recorded as the MIC value. Each experiment was repeated three times [30].

#### 4.2.2. Circular Dichroism (CD) of AOD

The secondary structure of AMPs can be determined by employing circular dichroism (CD) analysis on a MOS-450 spectropolarimeter (Bio-Logic, Seyssinet-Pariset, France). The CD spectra of the peptide were recorded in various solvents, including ddH_2_O, 20 mM SDS, and 50% TFE, utilizing a 1.0 mm optical path cuvette at ambient temperature. The wavelength range used was 180 nm to 260 nm, and each measurement was recorded three times [31].

#### 4.2.3. Drug Combination Index

The checkerboard method was employed to determine the antimicrobial activity of AOD in combination with tetracycline, ciprofloxacin, cephalosporin, and vancomycin against *S. epidermoides* G-81 [32]. AOD and antibiotics were added to a sterile 96-well plate in interleaved doses of 10 μL at concentrations of 8× MIC–1/16× MIC. The bacterial solution was adjusted to a concentration of 1.0 × 10^5^ CFU/mL after being transferred to the logarithmic growth phase. The 80 μL of the solution was added to each well of a 96-well plate, which was statically cultured at 37 °C for 16–24 h to determine the minimum inhibitory concentration (MIC) of the combination therapy. The combined effect was assessed by calculating the MIC ratio (FICI) for each combination. The FICI represents the ratio of the MIC value when the drug is used in combination with other drugs to its MIC value when used alone. The FICI is determined by adding the MIC value of the FIC peptide to that of the FIC antibiotic. The criteria are as follows: FICI ≥ 4 indicates antagonism when the two drugs are combined, 1 < FICI ≤ 4 is irrelevant, 0.5 < FICI ≤ 1 is additive action, and FICI ≤ 0.5 is synergistic action.

#### 4.2.4. AOD In Vivo Safety

ICR mice of similar body weight (6–8 weeks old) were carefully selected and intraperitoneally injected with AOD (10 mg/kg) on a daily basis for six consecutive days. Each group consisted of five mice. The changes in the mice’s body weight were recorded daily, while any behavioral abnormalities were closely observed. On the seventh day, blood samples were collected from the mice for subsequent blood and biochemical analysis. Simultaneously, liver and kidney samples were obtained from the mice for tissue fixation and HE staining to examine potential pathological changes.

### 4.3. Bactericidal Mechanism

#### 4.3.1. Fluorescence Staining

The *S. epidermidis* G-81 cultures, cultured overnight, were diluted to a concentration of 1 × 10^9^ CFU/mL. Subsequently, they were washed twice with 0.9% NaCl and incubated with AOD at concentrations of 1× MIC, 2× MIC, and 3× MIC for a duration of one hour. Following the incubation period, the samples were thoroughly cleaned and resuspended. Staining was carried out using SYTO9 and PI dyes for a duration of fifteen minutes at room temperature, followed by two to three washes. Finally, the stained samples were observed under a fluorescence microscope.

#### 4.3.2. Cell Membrane Fluidity

The logarithmic solution of *S. epidermidis* G-81 was diluted to an optical density (OD) of 1 at a wavelength of 630 nm. Subsequently, 10 μM Laurdan was added, and the solution was incubated at a temperature of 37 °C for a duration of 10 min. The bacterial solution was then subjected to two rounds of cleaning with phosphate-buffered saline (PBS) before being added to each well in a 96-well plate, with each well containing 90 μL. Next, different concentrations (1×, 2×, and 4× MIC AOD as well as 2× MIC lincomycin) were added to separate wells and incubated without exposure to light at a temperature of 37 °C for a period of 30 min. Each gradient had three replicates. The excitation wavelength was set at 490 nm, while the emission wavelength was set at 440 nm. PBS served as the blank control (CK), whereas glutaraldehyde functioned as the positive control. The luminescence signals were measured using the Infinite M200 Microplate reader (Tecan, Männedorf, Switzerland).

#### 4.3.3. Detection of Intracellular ATP Activity

*S. epidermidis* G-81 was activated overnight, cultured to a logarithmic stage, and then diluted to 5 × 10^8^ CFU/mL. The bacteria were incubated with 1×, 2×, and 4× MIC of AOD as well as 2× MIC of lincomycin at a temperature of 37 °C for a duration of one hour. Following centrifugation and precipitation, lysozyme was added, and the mixture was further incubated for an additional hour. The resulting supernatant was collected, and intracellular ATP changes were detected using an ATP detection kit provided by Beyotime Biotechnology Co. Ltd. (Shanghai, China).

#### 4.3.4. ROS Detection

*S. epidermidis* G-81 was cultured to the logarithmic stage and diluted to a concentration of 1 × 10^8^ CFU/mL. The DCFH-DA solution was added to the bacterial solution until the final concentration reached 10 μM. Subsequently, the mixture was incubated at 37 °C for 30 min and washed three times with PBS. Afterward, the mixture underwent another incubation at 37 °C for an additional 30 min. Next, the bacterial solution was incubated with 90 μL of DCFH-DA solution and mixed with AOD at concentrations of 1×, 2×, and 4× MIC, as well as lincomycin at a concentration of 2× MIC. Finally, an enzyme marker from Beyotime Biotechnology’s Reactive Oxygen Species Assay Kit was employed to detect the results.

### 4.4. Effects of AOD against S. epidermidis G-81 Biofilms

#### 4.4.1. Effects of AOD on Early Biofilm Formation

*S. epidermidis* G-81, which had reached the logarithmic stage, was diluted to 1 × 10^8^ CFU/mL and cultured in 96-well plates with 200 μL per well. The AOD and lincomycin were serially diluted twice in 24-well plates. The wells without antibacterial substances served as negative controls (CK), while the wells containing fresh TSB medium served as blank controls. The cultures were incubated at 37 °C for 24 h, followed by crystal violet staining. The supernatant was carefully removed from each well, and then the wells were washed three times with PBS and allowed to air dry. The fixing solution was then discarded, and the wells were washed twice with PBS. Subsequently, 100 μL of 2.5% glutaraldehyde was added to each well, and the solution was allowed to sit for 90 min. The wells were stained with 0.1% crystal violet (100 μL) for 15 min, rinsed with distilled water to remove excess dye solution, and left to dry at room temperature. Finally, any remaining dye was dissolved by adding 200 μL of 95% ethanol for 30 min before measuring the OD value at a wavelength of 570 nm.

#### 4.4.2. Effects of AOD on Mature Biofilm Formation

*S. epidermidis* G-81, which had reached the logarithmic stage, was diluted to 1 × 10^8^ CFU/mL and cultured in a 96-well plate at 37 °C for 24 h. AOD and lincomycin were added after gradient dilution, and the mixture was cultured in a 37 °C incubator for 24 h. The negative control (CK) was the hole without an antibacterial substance, and the blank control was the hole of the TSB medium. The crystal violet staining method was used as described above.

#### 4.4.3. Minimal Biofilm Inhibitory Concentration and Minimal Biofilm Eradication Concentration Determination

The anti-biofilm activity of AOD was assessed using the minimum biofilm inhibitory concentration (MBIC) and minimum biofilm eradication concentration (MBEC) methods [33]. MBIC is defined as the lowest concentration of AOD at which visible biofilm growth is not observed in a 96-well plate. The *S. epidermidis* G-81 was cultured until it reached the logarithmic growth phase, diluted to a concentration of 1 × 10^8^ CFU/mL, and incubated at 37 °C for 24 h. The upper layer planktonic cells were removed using PBS, and an AOD gradient diluent was prepared by incubating TSB (1–64× MIC) at 37 °C for 24 h. The lowest concentration without biofilm formation was recorded. MBEC is defined as the minimum concentration required to eradicate biofilm. After incubating the peptide and biofilm for 24 h, the biofilm was exposed to 100 μL of MTT solution (0.5 mg/mL) and incubated in the dark at 37 °C for 4 h. Subsequently, the MTT solution was replaced with 150 μL of DMSO solution, followed by measurement of absorbance at a wavelength of 550 nm. Negative controls were prepared using biofilms without AOD. The plate was then incubated at 37 °C for an additional 24 h before measuring its OD value.

#### 4.4.4. Biofilm Observation by Confocal Laser Scanning Microscopy

To validate the inhibitory and bactericidal effects of AOD on *S. epidermidis* G-81 biofilm and internal bacteria, the *S. epidermidis* G-81 was diluted to a concentration of 1 × 10^8^ CFU/mL and combined with AOD at concentrations of 1× MIC, 2× MIC, and 4× MIC. The mixture was then incubated at a temperature of 37°C for a duration of 24 h, while PBS served as the control (CK). In the resting cultures using a TSB medium, the bacteria formed biofilms on the slide surface. Following the incubation period, the suspended bacteria were gently rinsed twice with PBS, and the biofilm was stained using PI and SYTO9 from the LIVE/DEAD BacLight bacterial viability kit (ThermoFisher, Waltham, MA, USA) for a duration of 15 min. Subsequently, after rinsing the slides with PBS, the biofilm was observed under a Zeiss LSM880 confocal microscope (Carl Zeiss, Oberkochen, Germany). The SYTO9 stain has an excitation/emission wavelength of 480/500 nm, whereas PI has an excitation/emission wavelength of 490/635 nm [34].

#### 4.4.5. Effect of AOD on the Activity of Persistent Bacteria in Biofilm

*S. epidermidis* G-81 was cultured to the logarithmic stage, diluted to 1 × 10^8^ CFU/mL, and 200 μL of the bacterial solution was added to each well of a 24-well plate. The plate was incubated at 37 °C for 24 h, followed by removal of the floating bacteria through three washes with PBS. Subsequently, the bacteria were incubated with a TSB medium containing vancomycin at a concentration of 100 μg/mL for 24 h and then washed again with PBS three times. Persistent bacteria were obtained by adding AOD and lincomycin at concentrations ranging from 1–16× MIC, followed by incubation at 37 °C for 24 h. The number of viable bacteria was determined using the colony counting method [17].

## 5. Conclusions

In summary, it was found that the intact three-pair disulfide bond is crucial for both the secondary structure and antibacterial activity of AOD. Meanwhile, AOD can be utilized in conjunction with antibiotics to decrease the dosage. It caused deterioration in the fluidity of the membrane and led to metabolic disorders, resulting in the death of the bacterium. Furthermore, AOD demonstrated a significant inhibitory effect on both the initial and mature biofilms of *S. epidermidis* G-81, effectively eradicating intra-biofilm organisms by more than six orders of magnitude. This highlights its potential as a novel dual-functional agent with antibacterial and anti-biofilm properties.

## Figures and Tables

**Figure 1 marinedrugs-22-00463-f001:**
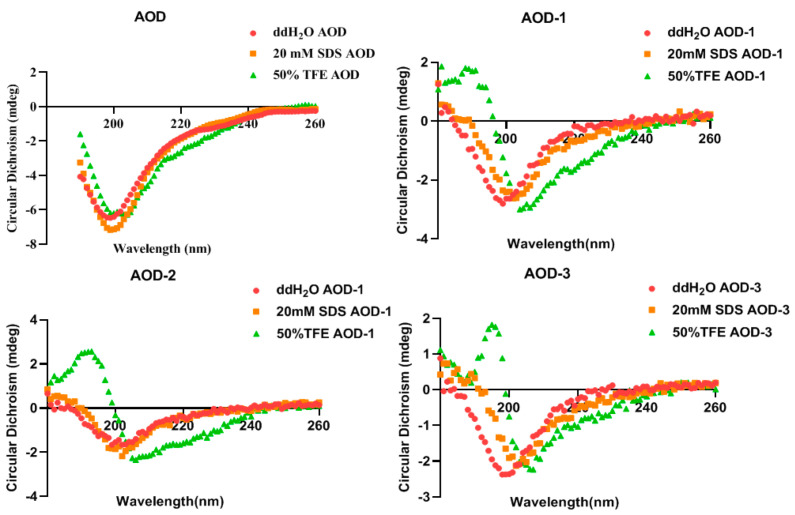
CD spectra of AOD and its derivatives.

**Figure 2 marinedrugs-22-00463-f002:**
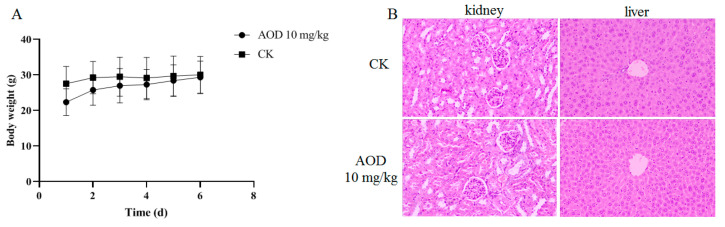
The safety of AOD in vivo. (**A**) Changes in body weight of mice within seven days. (**B**) Staining of liver and kidney tissue sections of mice.

**Figure 3 marinedrugs-22-00463-f003:**
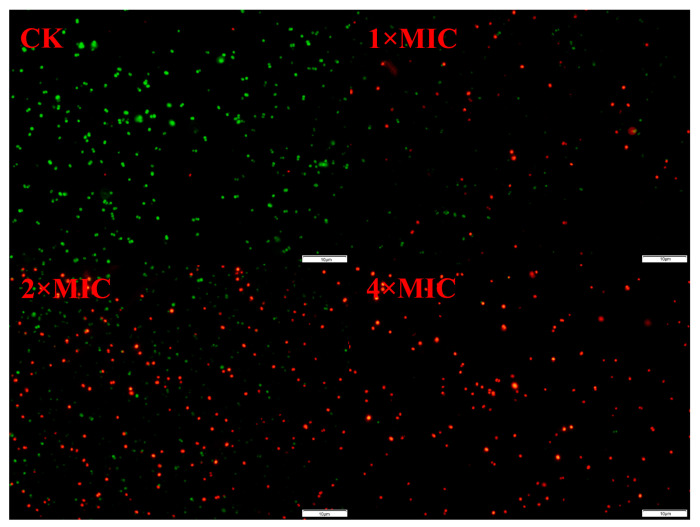
PI and SYTO9 staining of *S. epidermidis* G-81 treated by AOD.

**Figure 4 marinedrugs-22-00463-f004:**
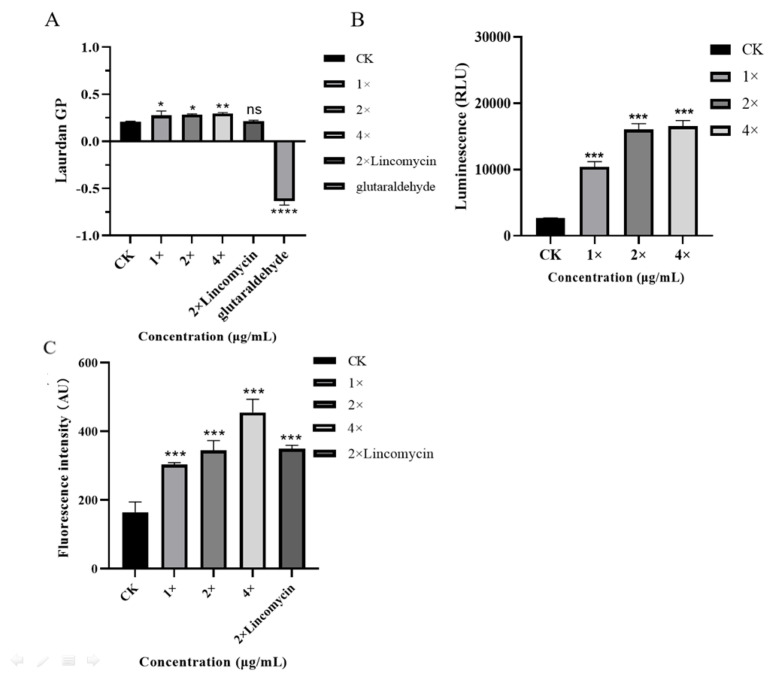
Effects of AOD on cell membrane fluidity (**A**), intracellular ATP (**B**), and reactive oxygen species (**C**). Results were given as mean ± SD (*n* = 3). ns: not significant, * *p* < 0.05, ** *p* < 0.01, *** *p* < 0.001, and **** *p* < 0.0001.

**Figure 5 marinedrugs-22-00463-f005:**
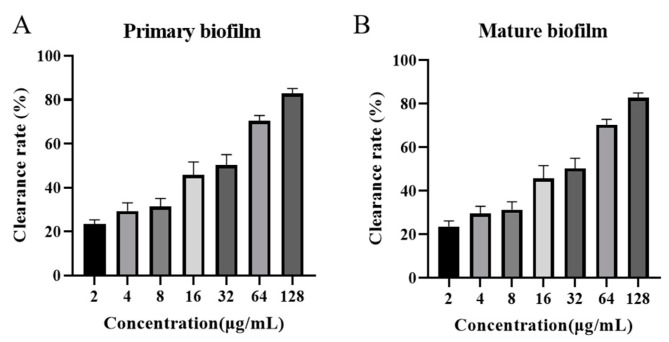
Effect of AOD on *S. epidermidis* G-81 primary (**A**) and mature (**B**) biofilms.

**Figure 6 marinedrugs-22-00463-f006:**
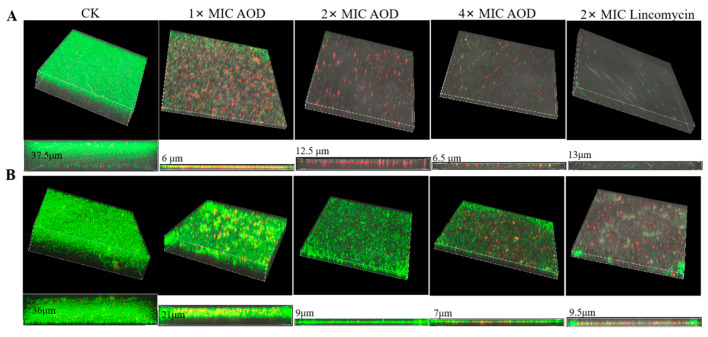
Effect of AOD on primary (**A**) and mature (**B**) biofilms observed by LCSM.

**Figure 7 marinedrugs-22-00463-f007:**
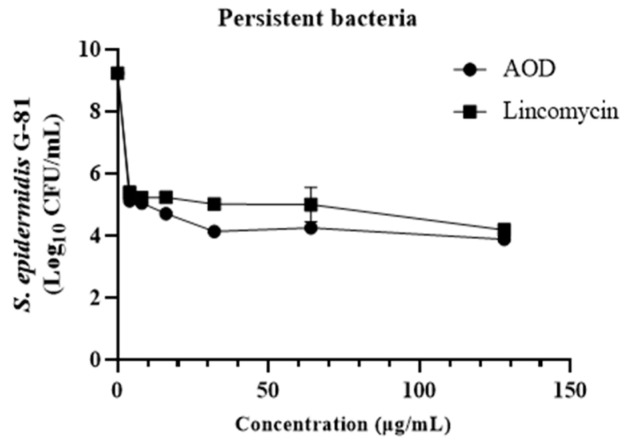
Effect of AOD on biofilm-retaining bacteria.

**Table 1 marinedrugs-22-00463-t001:** Sequences and physicochemical properties of AOD and its derivatives.

Peptides	Sequences	Disulfide Bonds	MW (Da)	PI	Charge	GRAVY
AOD	GFGCPWNRYQCHSHCRSIGRLGGYCAGSLRLTCTCYRS	C4-C25 C11-C33 C15-C35	4264.89	8.73	5	−0.363
AOD-1	GFGCPWNRYQCHSH**S**RSIGRLGGYCAGSLRLTCT**S**YRS	C4-C25 C11-C33	4234.79	9.55	5.5	−0.537
AOD-2	GFG**S**PWNRYQCHSHCRSIGRLGGY**S**AGSLRLTCTCYRS	C11-C33 C15-C35	4234.79	9.55	5.5	−0.537
AOD-3	GFGCPWNRYQ**S**HSHCRSIGRLGGYCAGSLRLT**S**TCYRS	C4-C25 C15-C35	4234.79	9.55	5.5	−0.537

MW: molecular weight; PI: isoelectric point; GRAVY: grand average of hydropathicity.

**Table 2 marinedrugs-22-00463-t002:** MIC of AOD and its derivatives.

Strains	MIC (μg/mL)				
	AOD	AOD-1	AOD-2	AOD-3	Lincomycin
*S. aureus* ATCC 43300	8	64	>64	>64	>64
*S. aureus* ATCC 25923	16	>64	>64	>64	1
*S. aureus* E48	8	64	>64	>64	1
*S. epidermidis* ATCC 12228	4	64	>64	>64	4
*S. epidermidis* G-81	4	64	>64	>64	4
*E. coli* ATCC 25922	>64	>64	>64	>64	2
*Salmonella typhimurium* CVCC 14028	>64	>64	>64	>64	>64
*Shigella flexneri* CMCC 51571	>64	>64	>64	>64	>64
*Pseudomonas aeruginosa* CICC 21625	>64	>64	>64	>64	>64

**Table 3 marinedrugs-22-00463-t003:** Secondary structure proportions of AOD and its derivative peptides.

Secondary Structure	H_2_O				20 mM SDS				50% TFE			
	AOD	AOD -1	AOD -2	AOD -3	AOD	AOD -1	AOD -2	AOD -3	AOD	AOD -1	AOD -2	AOD -3
Helix	6.18	5.82	6.03	5.8	7.23	6.2	6.47	6.37	8.51	7.8	8.14	7.5
Antiparallel	31.85	40.47	42.76	41.49	28.1	43.31	30.68	43.63	28.6	43.3	45.9	45.25
Parallel	5.11	3.26	3.41	3.35	5.3	3.44	5.09	3.47	5.09	3.71	3.93	3.7
Beta-Turn	22.2	18.56	17.64	18.19	23.17	17.71	23.1	17.66	23.1	17.89	16.65	17.27
Rndm.Coil	34.5	31.79	30.16	31.17	36.2	29.34	34.7	28.86	34.7	27.3	25.38	26.28

**Table 4 marinedrugs-22-00463-t004:** AOD in combination with antibiotics against *S. epidermidis* G-81.

Combination	Variety	*S. epidermidis* G-81			
		MIC_a_ (μg/mL)	MIC_c_ (μg/mL)	FIC	FICI
AOD-CEF	AOD	4	2	0.5	0.575
	CEF	1	0.075	0.075	
AOD-TC	AOD	4	2	0.5	1
	TC	2	1	0.5	
AOD-CIP	AOD	4	0.5	0.125	1.125
	CIP	2	2	1	
AOD-VAN	AOD	4	2	0.5	0.581
	VAN	4	0.125	0.031	

MIC_a_: single agent MIC; MIC_c_: combined agent MIC; TC: tetracycline; CIP: ciprofloxacin; CEF: ceftiofur; VAN: vancomycin. FICI = (MIC of drug A in combination)/(MIC of drug A alone) + (MIC of drug B in combination)/(MIC of drug B alone). FICI ≤ 0.5 indicates a synergistic effect, 0.5 < FICI ≤ 1 indicates an additive effect, 1 < FICI ≤ 4 indicates an irrelevant effect, and FICI > 4 indicates an antagonistic effect.

**Table 5 marinedrugs-22-00463-t005:** Hematological indices in mice after one week of AOD treatment.

Parameter	Unit	Control	AOD 10 mg/kg
WBC	10^9^ cells/mL	7.96 ± 0.22	8.56 ± 0.35
NEUT	10^9^ cells/mL	1.46 ± 0.069	1.69 ± 0.14
LYM	10^9^ cells/mL	5.96 ± 0.14	6.24 ± 0.16
MONO	10^9^ cells/mL	0.26 ± 0.017	0.25 ± 0.035
EO	10^9^ cells/mL	0.18 ± 0.01	0.20 ± 0.001
BASO	10^9^ cells/mL	0.13 ± 0.01	0.16 ± 0.01
NEUT	%	18.33 ± 0.38	19.71 ± 0.75
LYM	%	74.86 ± 0.44	73.13 ± 0.01
MONO	%	2.85 ± 0.15	2.94 ± 0.28
EO	%	2.29 ± 0.09	2.35 ± 0.01
BASO	%	1.68 ± 0.07	1.86 ± 0.08
RBC	10^12^/L	9.11 ± 0.12	9.835 ± 0.13
HBG	g/L	111.75 ± 1.44	127.25 ± 1.75
HCT	%	39.4 ± 0.56	43.025 ± 0.0611
MCV	fL	40.1 ± 0.204	40.55 ± 0.26
MCH	pg	12.8 ± 0.041	12.925 ± 0.047
MCHC	g/L	32.4 ± 0.125	32.075 ± 0.25
PLT	10^9^/L	826.25 ± 9.42	811 ± 5.64
MPV	fL	6.425 ± 0.025	6.475 ± 0.047
PDW	%	12.425 ± 0.048	16.1 ± 0.13
PLCR	%	12.425 ± 0.66	16.15 ± 0.51

**Table 6 marinedrugs-22-00463-t006:** MBIC and MBEC of AOD on *S. epidermidis* G-81.

Strain	MBIC (μg/mL)		MBEC (μg/mL)	
	AOD	Lin	AOD	Lin
*S. epidermidis* G-81	16	64	32	>128

Lin: lincomycin.

## Data Availability

The original contributions presented in the study are included in the article; further inquiries can be directed to the corresponding authors.

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
