# Peer review of "The Marine Antimicrobial Peptide AOD with Intact Disulfide Bonds Has Remarkable Antibacterial and Anti-Biofilm Activity"

_marinedrugs, 2024, doi:10.3390/md22100463_

Round 1

Reviewer 1 Report

Comments and Suggestions for Authors

The authors investigated the bioactivity and toxicity of American Oyster Defensin (AOD) peptide against a biofilm.  They demonstrated various experiment tools that can be used to quantize an AMP’s activity against the biofilm and tested also the activity upon deleting AOD’s disulfide bond. I have a few questions/comments on this work:

1.      The title stresses the disulfide bonds are essential for the AMP structure and anti-biofilm activity.  However, it is known that disulfide bonds are crucial for maintaining protein or peptide structures, which are often crucial for their activities as well. Thus, table 2 does not bring any new concept but merely confirmed this knowledge. Furthermore, the authors provide CD spectrum (Figure 1) to show how the helix content is affected by the disulfide bond, but preferably one should show the change in structure (e.g. NMR or at least AlphaFold structure).  It is a pity that they do not dig deeper to reveal more insights.  Except for Table 2 and Figure 1, all experiments are performed using AOD (native, all disulfide bonds preserved).  Thus, the focus on the paper seems to be AOD, rather than the disulfide bonds.  I would recommend the authors find a more suitable title that reflects the points of this work, e.g. activity against biofilm, toxicity test, etc.

2.      Some abbreviations appeared without explanation.  For instance, FICI is never fully spelled. The authors are recommended to go through all abbreviations again and make sure that the terms are defined.

Author Response

Comment 1:The title stresses the disulfide bonds are essential for the AMP structure and anti-biofilm activity.  However, it is known that disulfide bonds are crucial for maintaining protein or peptide structures, which are often crucial for their activities as well. Thus, table 2 does not bring any new concept but merely confirmed this knowledge. Furthermore, the authors provide CD spectrum (Figure 1) to show how the helix content is affected by the disulfide bond, but preferably one should show the change in structure (e.g. NMR or at least AlphaFold structure).  It is a pity that they do not dig deeper to reveal more insights.  Except for Table 2 and Figure 1, all experiments are performed using AOD (native, all disulfide bonds preserved). Thus, the focus on the paper seems to be AOD, rather than the disulfide bonds.  I would recommend the authors find a more suitable title that reflects the points of this work, e.g. activity against biofilm, toxicity test, etc.

Answer 1: Thank you for your comments. The title of this manuscript has been revised as “The marine antimicrobial peptide AOD with intact disulfide bonds has remarkable antibacterial and anti-biofilm activity”.

Comment 2: Some abbreviations appeared without explanation.  For instance, FICI is never fully spelled. The authors are recommended to go through all abbreviations again and make sure that the terms are defined.

Answer:Thanks for your comments, FICI is fractional inhibitory concentration index, FICI = (MIC of drug A in combination) / (MIC of drug A alone) + (MIC of drug B in combination) / (MIC of drug B alone), has been supplemented in the table notes of Table 3, and relevant notes have been added to the text. At the same time, abbreviations have been checked and notes added throughout the text.

Reviewer 2 Report

Comments and Suggestions for Authors

The authors present an interesting work in which they evaluate the impact of disulfide bonds on the activity of the antimicrobial peptide, American oyster defensin (AOD). The antimicrobial activity in strains of S. epidermidis, S. aureus and E. coli, show that the parent peptide maintains the activity with respect to the mutants called AOD1, AOD2 and AOD3. In each of these mutants, one disulfide bond was eliminated, leaving only two of the three that the parent peptide presents. These changes were associated with function and by CD they observed signals that suggest the loss of alpha helix structure in the mutants in an environment that simulates the membrane using 50% TFE. Their results suggest that the presence of the three disulfide bonds maintains the alpha helix structure that seems to be fundamental for antimicrobial activity. In addition, they evaluated the changes in the MIC of parental AOD when applied together with other antibiotics on S. epidermidis. They also examined the in vivo safety of parental AOD in mice. And finally they evaluated the inhibitory effect of parental AOD on biofilm formation.

I only have a few observations that I hope will be useful to the authors:

1.- Abstract: I believe the abstract could be more concise. While the comparison with lincomycin is emphasized, it would be beneficial to also highlight the reduced efficacy of the mutants compared to the parent peptide, providing specific data. The comparison between mutant and parent peptides seems somewhat minimized in this section.

2.- Line 78-80: The disulfide bond is a prevalent and crucial structural and functional moiety found in defensins, playing a pivotal role in the antibacterial activity, proper folding, and spatial stability of defensin molecules.

It could be summarized as: The disulfide bond is a common and essential structural element in defensins, playing a key role in their antibacterial activity, proper folding, and structural stability.

3.- Introduction: I recommend including at least one additional example of defensins where cysteine residues are mutated or depleted and how this affects the structure-function relationship. Although examples are mentioned in the discussion, having one in the introduction would clarify the significance of mutating cysteine residues.

4.- In lines 97-99 (results): The cysteines were exchanged by serines because it is proved to be the attractive substitutable residue with the similar chemical composition and structure compared to cysteines.

How did they demonstrate this? If it is a previous work, put the reference. If the authors determined it, say how they did it.

5.- AOD1 Activity: How do the authors relate AOD1 activity to changes in cysteine residues? Does serine contribute to maintaining antimicrobial activity? How do these changes correlate with the CD spectrum? I believe this aspect requires further discussion.

6.-  If the GRAVY index, isoelectric point and charge probably made the mutants more hydrophilic, how does this alter the structure and function? Could you speculate something and mention it in the discussion?

7.- It is unclear why lincomycin was used in the study. Please provide an explanation.

8.- Your results are very interesting but it would have been interesting to test other antibiotics with a different mechanism of action that would also allow you to eliminate biofilms.

9.- "Lines 188-190: The findings suggest that AOD triggers the generation of reactive oxygen species in S. epidermidis G-81 cells, resulting in enhanced cellular apoptosis and diminished development of drug resistance". Add a reference to support the last statement or write it as a probable event.

10.- Could you improve the distribution of sections A, B and C of graph 4?

11.- How do you associate the low activity of AOD with E. coli? Why did you not evaluate more gram-negative strains?

12.- Your results show that AOD has a dose-dependent effect on reducing the thickness of the mature film, but this is not observed in the primary films. What could be the reason for this?

13.- Please, could you define Abu when it appears in the text: alpha-amino-n-butyric acid (Abu).

Author Response

Comments 1.- Abstract: I believe the abstract could be more concise. While the comparison with lincomycin is emphasized, it would be beneficial to also highlight the reduced efficacy of the mutants compared to the parent peptide, providing specific data. The comparison between mutant and parent peptides seems somewhat minimized in this section.

Answer 1:Thanks to your comments, we have revised the abstract section to add the results of the parent peptide compared to the mutant and to streamline it.

Comments 2.- Line 78-80: The disulfide bond is a prevalent and crucial structural and functional moiety found in defensins, playing a pivotal role in the antibacterial activity, proper folding, and spatial stability of defensin molecules.

It could be summarized as: The disulfide bond is a common and essential structural element in defensins, playing a key role in their antibacterial activity, proper folding, and structural stability.

Answer 2:Thank you for your comments, the relevant sections have been revised.

Comments 3.- Introduction: I recommend including at least one additional example of defensins where cysteine residues are mutated or depleted and how this affects the structure-function relationship. Although examples are mentioned in the discussion, having one in the introduction would clarify the significance of mutating cysteine residues.

Answer 3: Thank you for your suggestion. There are two examples to evaluate the effects of cysteine residues to the activities and structure of antimicrobial peptides, which showed as follows: “. It has been reported that coprisin from Copris tripartitus, consisting of 43 amino acids, lacked any disulphide bonds pairs, thereby attenuating its antibacterial activity while still retaining its antifungal activity [10]. The linear hBD-3-l has been demonstrated to exhibit the highest activity, whereas the stability of hBD-3 against protease activity diminishes as the number of disulfide bonds decreases [11].”

Comments 4.- In lines 97-99 (results): The cysteines were exchanged by serines because it is proved to be the attractive substitutable residue with the similar chemical composition and structure compared to cysteines.

How did they demonstrate this? If it is a previous work, put the reference. If the authors determined it, say how they did it.

Answer 4:Thank you for your comments, relevant references have been added.

Comments 5.- AOD1 Activity: How do the authors relate AOD1 activity to changes in cysteine residues? Does serine contribute to maintaining antimicrobial activity? How do these changes correlate with the CD spectrum? I believe this aspect requires further discussion.

Answer 5: Thank you for the comments. It has been revised in the section of discussion as following: “The AOD exhibiting complete three-pair disulfide bonds demonstrated the highest level of antimicrobial activity, and exhibited the greatest abundance of α-helical structures in TFE solution. The substitutions of cysteine with serine in the mutants lead to a substantial decrease in its antibacterial activity. Simultaneously, a concurrent decline is observed in the proportion of α-helix formation when exposed to SDS and TFE solutions (Figure 1 and Table 3). The substitution of cysteine as serine, therefore, exerts a profound influence on both the antibacterial efficacy and structural characteristics of AOD.”

Comments 6.-  If the GRAVY index, isoelectric point and charge probably made the mutants more hydrophilic, how does this alter the structure and function? Could you speculate something and mention it in the discussion?

Answer 6: Thank you for the question. It has been revised in the section of discussion as following: “Positive net charge and hydrophilicity are the two main factors required for the antimicrobial activity of AMPs [14]. The GRAVY is the grand average of hydropathicity. AMPs bind to the surface of bacteria through electrostatic bonding, which is thought to be the first step in promoting the interaction between AMPs and cell membranes. In this work, the GRAVY index of AOD-1, 2, 3 decreased from -0.363 to -0.573, indicating that the derivatives were more hydrophilic, which reduces its binding to the surface of the bacterial cell membrane and results in a decrease in activity.”

Comments 7.- It is unclear why lincomycin was used in the study. Please provide an explanation.

Answer 7:Thank you for your question, this article is based on a previous article (High yield preparation of American Oyster Defensin (AOD) via a small and acidic fusion tag and its functional characterization) In the previous study, we verified the protective effect of AOD on mice infected with Staphylococcus epidermidis G-81, and lincomycin, which is commonly used in clinical treatment, was chosen as the control group. This is because lincomycin is more commonly used in the treatment of skin diseases and it has a similar antibacterial spectrum with AOD. The use of lincomycin as the control group was continued, resulting in a consistent approach.

Comments 8.- Your results are very interesting but it would have been interesting to test other antibiotics with a different mechanism of action that would also allow you to eliminate biofilms.

Answer 8: Thank you for the comments. In this study, it has been shown that AOD resulted in the impairment of membrane fluidity and induced metabolic disorders, ultimately leading to bacterial death. Furthermore, it was demonstrated that AOD exhibited a notable inhibitory impact on the biofilm. Meanwhile, the lincomycin with similar antibacterial spectrum was used as control in the antimicrobial and anti-biofilm. The mechanism of lincomycin is different from AOD. It is exerted on the bacterial ribosome by binding to the central ring of the 23S rRNA gene within the 50S subunit, thereby impeding elongation of peptide chains and effectively inhibiting protein synthesis in bacterial cells. Other antibiotics with a different mechanism of action may be used and study in our further study.

Comments 9.- "Lines 188-190: The findings suggest that AOD triggers the generation of reactive oxygen species in S. epidermidis G-81 cells, resulting in enhanced cellular apoptosis and diminished development of drug resistance". Add a reference to support the last statement or write it as a probable event.

Answer 9:Thank you for your comments, relevant references have been added.

Comments 10.- Could you improve the distribution of sections A, B and C of graph 4?

Answer 10:Thank you for your comments, the distribution of Figure 4 have been improved as follows:

Comments 11.- How do you associate the low activity of AOD with E. coli? Why did you not evaluate more gram-negative strains?

Answer 11: Thank you for your question. The native AOD purified from American oyster displays a broad antibacterial spectrum against both gram-positive and negative bacteria with the minimal effective concentration(MECs) of 10 and 32 μg/mL to S. aureus and E. coli, respectively (DOI: 10.1016/j.bbrc.2005.11.013). Meanwhile, in our previous work, there was no antimicrobial activity of AOD against gram-negative bacteria (DOI: 10.3390/md22010008). This contradiction may be due tothe differences in measurement methods between MEC (DOI: 10.1016/j.bbrc.2005.11.013) and MIC (DOI: 10.3390/md22010008). The activity of AOD to more gram-negative strains including Salmonella typhimurium CVCC 14028, Shigella flexneri CMCC 51571 and Pseudomonas aeruginosa CICC 21625 have been added and shown in Table 2. Consistent with previous findings, no antimicrobial activity was observed against these gram-negative bacteria.

Comments 12.- Your results show that AOD has a dose-dependent effect on reducing the thickness of the mature film, but this is not observed in the primary films. What could be the reason for this?

Answer 12: Thank you for the comments. It has been reconfirmed that the antibacterial peptide AOD exhibits no dosage-dependent impact on reducing the thickness of the primary films. This may due to the simple composition of the initial biofilms, mainly involving the surface proteins and adhesions of bacterial. And the antimicrobial peptide AOD has no effect on these two elements. In the contrary, the mature biofilm model is composed of an outer layer representing the primary biofilm, an intermediate layer serving as a linker, an inner layer designated for the conditioner, and a matrix layer. And the composition of a mature biofilm is intricate, primarily comprising polysaccharide-protein complexes secreted by bacteria, a polysaccharide matrix, fibrin, lipoproteins, and additional polysaccharide-protein complexes. The action of AOD on one or more constituent substances in mature biofilms will directly result in the disruption of the membrane structure, thereby inducing a dose-dependent effect. Further study will be conducted to elucidate additional details regarding the mechanism of action.

Comments 13.- Please, could you define Abu when it appears in the text: alpha-amino-n-butyric acid (Abu).

Answer 13: Thank you for your question. The full name of Abu (alpha-amino-n-butyric acid) has been added in the manuscript when it first appeared.

Reviewer 3 Report

Comments and Suggestions for Authors

The study by Mao and colleagues shows the structure-function relationship of an antimicrobial peptide AOD. They showed that disulfide bonds are required for AOD to exhibit antimicrobial activity against the strains of Gram-positive bacteria. Removal of disulfide bonds by substituting cysteine residues with serine residues abolished the antimicrobial activity as well as disrupted the alpha-helical structure. AOD kills the bacteria through a membrane lytic mechanism. It also inhibits biofilm formation and persistent bacteria.

Minor comments:

  1. In lines 96-97, Cysteine substitution with Serine needs to be written properly for AOD mutants 2 and 3.
  2. In Table 1, it would be better if the authors could highlight the cysteine residues and substituted cysteine with serine residues so that people can easily understand the positions of cysteine and serine residues.
  3. Authors did not include the observed molecular weight of the peptides.
  4. Write about the statistics test(s) in legends where you have used stars to show the significance.
  5. In Fig 5, the y-axis should show the values in percentage.
  6. In the method section, what kind of plates (plate material) are used for biofilm formation needs to be mentioned.

Major comments:

  1. AOD is Gram-positive bacteria specific. It would have been better if the authors had included a few more Gram-negative bacterial strains to confirm the specificity of AOD against Gram-positive bacteria firmly.
  2. It looks like the interpretation of CD data is not proper. In lines 117-118, the authors wrote that AOD takes α and β secondary structures in all three environments, i.e., water, SDS, and TFE. But it’s not depicted in Figure 1a. In water, it shows a negative peak/band at 200 nm, meaning AOD behaves like an unstructured peptide.

In the SDS environment also, it looks almost the same as water, with no recognizable difference.

In TFE, it slightly shifted towards 208 nm but not at 208 nm; it looked around 203 nm. It might have some percentage of a helical structure. There are no clear negative peaks at 208 and 222 nm in all three environments.

And why the authors claimed that AOD shows β structure also (lines 117-118). It’s a misinterpretation. In any environment, there is no negative band at 217 nm.

The mutants AOD-1, -2, and -3 all show comparable structures to AOD in water and SDS environments. Even all the mutants show slightly better α helical properties in the TFE environment than AOD (better-shaped negative peaks at 208 and 222 nm).

Regarding CD data, the authors need to revisit their observations.

3. SDS is not a true mimic of the bacterial membrane. To show the peptide structure in bacterial mimetic environment, CD spectra must be recorded in PE/PG or PC/PG lipid vesicles. (Ref PMID: 28051162)

In parallel, it would be better if authors could show CD spectra in a mammalian mimetic membrane environment (PC/Cholesterol lipid vesicles). Based on this, the toxicity profile can also be predicted. Nontoxic peptides should not adopt any structure in a mammalian mimetic membrane system.

Comments on the Quality of English Language

In the title, add an article 'an' before 'antimicrobial peptide'. Then the title will be ...'an antimicrobial peptide AOD'

Author Response

Minor comments:

Comments 1: In lines 96-97, Cysteine substitution with Serine needs to be written properly for AOD mutants 2 and 3.

Answer 1Thank you for your comments, it has been revised and checked.

Comments 2: In Table 1, it would be better if the authors could highlight the cysteine residues and substituted cysteine with serine residues so that people can easily understand the positions of cysteine and serine residues.

Answer 2: Thank you for your comments, we are already highlighting the replacement sites. 

Comments 3: Authors did not include the observed molecular weight of the peptides.

Answer 3: Thank you for your question. The molecular weight of the peptides were calculated by DBAASP (https://www.dbaasp.org/tools?page=property-calculation) and showed in Table 1.

Comments4: Write about the statistics test(s) in legends where you have used stars to show the significance.

Answer 4: Thank you for your comments, they have been added in the figure notes to Figure 4. 

Comments 5: In Fig 5, the y-axis should show the values in percentage.

Answer 5: Thank you for your comments, it has been revised as follows.

Comments 6: In the method section, what kind of plates (plate material) are used for biofilm formation needs to be mentioned.

Answer 6:Thank you for your comments, the plate material have been added to the material approach.

Major comments:

Comments 7: AOD is Gram-positive bacteria specific. It would have been better if the authors had included a few more Gram-negative bacterial strains to confirm the specificity of AOD against Gram-positive bacteria firmly.

Answer 7: Thank you for your question. The native AOD purified from American oyster displays a broad antibacterial spectrum against both gram-positive and negative bacteria with the minimal effective concentration(MECs) of 10 and 32 μg/mL to S. aureus and E. coli, respectively (DOI: 10.1016/j.bbrc.2005.11.013). Meanwhile, in our previous work, there was no antimicrobial activity of AOD against gram-negative bacteria (DOI: 10.3390/md22010008). This contradiction may be due to the differences in measurement methods between MEC (DOI: 10.1016/j.bbrc.2005.11.013) and MIC (DOI: 10.3390/md22010008). The activity of AOD to more gram-negative strains including Salmonella typhimurium CVCC 14028, Shigella flexneri CMCC 51571 and Pseudomonas aeruginosa CICC 21625 have been added and shown in Table 2. Consistent with previous findings, no antimicrobial activity was observed against these gram-negative bacteria.

Comments 8: It looks like the interpretation of CD data is not proper. In lines 117-118, the authors wrote that AOD takes α and β secondary structures in all three environments, i.e., water, SDS, and TFE. But it’s not depicted in Figure 1a. In water, it shows a negative peak/band at 200 nm, meaning AOD behaves like an unstructured peptide.

In the SDS environment also, it looks almost the same as water, with no recognizable difference.

In TFE, it slightly shifted towards 208 nm but not at 208 nm; it looked around 203 nm. It might have some percentage of a helical structure. There are no clear negative peaks at 208 and 222 nm in all three environments.

And why the authors claimed that AOD shows β structure also (lines 117-118). It’s a misinterpretation. In any environment, there is no negative band at 217 nm.

The mutants AOD-1, -2, and -3 all show comparable structures to AOD in water and SDS environments. Even all the mutants show slightly better α helical properties in the TFE environment than AOD (better-shaped negative peaks at 208 and 222 nm).

Regarding CD data, the authors need to revisit their observations.

Answer 8: Yes, you are right. We have rechecked the results and execute the quantitative evaluation of secondary structure from the CD spectrum via the program CDNN version 2.1, which you can see in the following table. Additionally, AOD, as the parental peptide, displayed the most prominent negative peaks (Figure 1) and had the highest proportion of helical structure (Table 3).

Table 3 Secondary structure proportions of AOD and its derivative peptides

Secondary structure

H2O

20mM SDS

50% TFE

AOD

AOD

-1

AOD

-2

AOD

-3

AOD

AOD

-1

AOD

-2

AOD

-3

AOD

AOD

-1

AOD

-2

AOD

-3

helix

6.18

5.82

6.03

5.8

7.23

6.2

6.47

6.37

8.51

7.8

8.14

7.5

Antiparallel

31.85

40.47

42.76

41.49

28.1

43.31

30.68

43.63

28.6

43.3

45.9

45.25

Parallel

5.11

3.26

3.41

3.35

5.3

3.44

5.09

3.47

5.09

3.71

3.93

3.7

Beta-Turn

22.2

18.56

17.64

18.19

23.17

17.71

23.1

17.66

23.1

17.89

16.65

17.27

Rndm.Coil

34.5

31.79

30.16

31.17

36.2

29.34

34.7

28.86

34.7

27.3

25.38

26.28

Comments 9: SDS is not a true mimic of the bacterial membrane. To show the peptide structure in bacterial mimetic environment, CD spectra must be recorded in PE/PG or PC/PG lipid vesicles. (Ref PMID: 28051162).

In parallel, it would be better if authors could show CD spectra in a mammalian mimetic membrane environment (PC/Cholesterol lipid vesicles). Based on this, the toxicity profile can also be predicted. Nontoxic peptides should not adopt any structure in a mammalian mimetic membrane system.

Answer 9: Thank you for your suggestion. As you have stated, SDS is not a true mimic of the bacterial membrane. However, due to the unique properties of SDS micelles, they can effectively serve as a model for bacterial cell membranes. As stated in the reference (DOI:10.1002/bip.20397) “SDS is a suitable mimic for the negatively charged molecules found in bacterial membranes. Although there is no direct correlation between the SDS head group and the head groups commonly found on the phospholipids composing a bacterial membrane (as there is between DPC and PC phospholipids), the SDS micelle is generally agreed to be a good model bacterial membrane because it possesses an anionic exterior and a hydrophobic interior.” Meanwhile, there are numerous references that utilize SDS as a bacterial cell membrane mimic for various analytical techniques, CD and NMR (DOI:10.1021/acs.jpcb.2c05909, DOI: 10.1021/jm701604t). Additionally, the cytotoxicity tests have been executed in our previous work (DOI:10.3390/md22010008), the results demonstrate that AOD is non-toxic characteristic (survival rate of Hacat cells: 128 µg/mL of AOD was 98%; the hemolytic rate: 256 µg/mL of AOD was 1.29%).

Round 2

Reviewer 2 Report

Comments and Suggestions for Authors

The authors have addressed all the questions and comments made to the first version, including adding bioassays. I consider that the paper has been improved and can be accepted in its current form for publication.

Author Response

Comments 1: The authors have addressed all the questions and comments made to the first version, including adding bioassays. I consider that the paper has been improved and can be accepted in its current form for publication.

Answer 1: I am extremely grateful for your review comments and suggestions. I will keep on modifying and perfecting the manuscript. 

Reviewer 3 Report

Comments and Suggestions for Authors

Comment 1: In lines 99-100, Cysteine substitution with Serine needs to be corrected for AOD mutants 2 and 3.

Comment 2: I suggested to include observed weight. There is a difference between the observed weight and the calculated weight. Calculated weight represents the theoretical weight, whereas observed weight is the actual/practical weight measured after synthesizing the peptide by MALDI-TOF or other methods. If they got synthesized peptide from a company, authors can ask for observed mol weight.

Comment 3: The p-values in Figure 4 legend are written incorrectly. It should be as *p < 0.05, **p < 0.01, ***p < 0.001, and ****p < 0.0001.

According to the authors, SDS represents a bacterial membrane environment. What does TFE represent? Is it a mammalian mimetic membrane? It’s a little unclear. In the text (line 119), it is written ‘microbial membrane.’ Are TFE and SDS both representing microbial/bacterial membranes?

Comments on the Quality of English Language

No issues.

Author Response

Comment 1: In lines 99-100, Cysteine substitution with Serine needs to be corrected for AOD mutants 2 and 3.

Answer 1: Thank you for your question. The cysteine substitutions with serine have been revised as "AOD-1 (C15, C35 →S15, S35), AOD-2 (C4, C25 →S4, S25), AOD-3 (C11, C33 →S11, S33)".

Comment 2: I suggested to include observed weight. There is a difference between the observed weight and the calculated weight. Calculated weight represents the theoretical weight, whereas observed weight is the actual/practical weight measured after synthesizing the peptide by MALDI-TOF or other methods. If they got synthesized peptide from a company, authors can ask for observed mol weight.

Answer 2: Thank you for your suggestion.  The report of the weight of AOD-1, AOD-2, and AOD-3 is showed as follows and the description has been revised in the manuscript:

AOD-1

AOD-2

AOD-3

Comment 3: The p-values in Figure 4 legend are written incorrectly. It should be as *p < 0.05, **p < 0.01, ***p < 0.001, and ****p < 0.0001.

Answer 3: Thank you for your suggestion. It has been revised as "*p < 0.05, **p < 0.01, ***p < 0.001, and ****p < 0.0001." in the manuscript.

Comments 4: According to the authors, SDS represents a bacterial membrane environment. What does TFE represent? Is it a mammalian mimetic membrane? It’s a little unclear. In the text (line 119), it is written ‘microbial membrane.’ Are TFE and SDS both representing microbial/bacterial membranes?

Answer 4: Thank you for your question. Yes, the TFE and SDS both representing microbial TFE and SDS both representing microbial membranes and it has been revised in the manuscript.